# Genetic Diversity, Population Structure and Linkage Disequilibrium Analyses in Tropical Maize Using Genotyping by Sequencing

**DOI:** 10.3390/plants11060799

**Published:** 2022-03-17

**Authors:** Bhupender Kumar, Sujay Rakshit, Sonu Kumar, Brijesh Kumar Singh, Chayanika Lahkar, Abhishek Kumar Jha, Krishan Kumar, Pardeep Kumar, Mukesh Choudhary, Shyam Bir Singh, John J. Amalraj, Bhukya Prakash, Rajesh Khulbe, Mehar Chand Kamboj, Neeraja N. Chirravuri, Firoz Hossain

**Affiliations:** 1ICAR-Indian Institute of Maize Research, Ludhiana 141004, India; bhupender.icar@gmail.com (B.K.); nagarsonu72@gmail.com (S.K.); brijeshsingh714@gmail.com (B.K.S.); chayanika.iimr@gmail.com (C.L.); abhishekimhuman@gmail.com (A.K.J.); krishjiwra@gmail.com (K.K.); pardeep.kumar656@gmail.com (P.K.); mukesh.agri08@gmail.com (M.C.); singhsb1971@rediffmail.com (S.B.S.); 2Centre for Plant Breeding and Genetics, Tamil Nadu Agricultural University, Coimbatore 641003, India; jnjoel@gmail.com; 3ICAR-Directorate of Poultry Research, Hyderabad 500030, India; drbhukyaprakash@gmail.com; 4Department of Crop Imrovement, ICAR-Vivekananda Parvatiya Krishi Anusandhan Sansthan, Almora 263601, India; rkkhulbe@gmail.com; 5Department of Plant Breeding, CCS-Haryana Agricultural University, Regional Research Station, Uchani 132001, India; kambojmehar@gmail.com; 6Department of Crop Improvement, ICAR-Indian Institute of Rice Research, Hyderabad 500030, India; cnneeraja@gmail.com; 7Division of Genetics, ICAR-Indian Agricultural Research Institute, New Delhi 110012, India; fh_gpb@yahoo.com

**Keywords:** genetic purity, genotyping-by-sequencing, population structure, linkage disequilibrium, haplotypes blocks, tropical maize

## Abstract

Several maize breeding programs in India have developed numerous inbred lines but the lines have not been characterized using high-density molecular markers. Here, we studied the molecular diversity, population structure, and linkage disequilibrium (LD) patterns in a panel of 314 tropical normal corn, two sweet corn, and six popcorn inbred lines developed by 17 research centers in India, and 62 normal corn from the International Maize and Wheat Improvement Center (CIMMYT). The 384 inbred lines were genotyped with 60,227 polymorphic single nucleotide polymorphisms (SNPs). Most of the pair-wise relative kinship coefficients (58.5%) were equal or close to 0, which suggests the lack of redundancy in the genomic composition in the majority of inbred lines. Genetic distance among most pairs of lines (98.3%) varied from 0.20 to 0.34 as compared with just 1.7% of the pairs of lines that differed by <0.20, which suggests greater genetic variation even among sister lines. The overall average of 17% heterogeneity was observed in the panel indicated the need for further inbreeding in the high heterogeneous genotypes. The mean nucleotide diversity and frequency of polymorphic sites observed in the panel were 0.28 and 0.02, respectively. The model-based population structure, principal component analysis, and phylogenetic analysis revealed three to six groups with no clear patterns of clustering by centers-wise breeding lines, types of corn, kernel characteristics, maturity, plant height, and ear placement. However, genotypes were grouped partially based on their source germplasm from where they derived.

## 1. Introduction

Maize (*Zea mays* L.) is one of the most valuable crops cultivated in the world for its diverse uses in food, feed, starch, textile, pharmaceutical, cosmetic industries, oil, protein, alcoholic, and bio-ethanol production [1,2,3,4]. In India, maize is the thirdmostmajor crop after rice and wheat and contributes 2–3% of the total maize produced globally [5]. It is one of the highest-yielding crops, which is primarily due to heterosis. The use of heterosis in crop improvement has been the major significant scientific achievement in the history of plant breeding, which has contributed a lot to an increase in yield and productivity [6,7].Heterosis can be exploited better in hybrids when well-characterized inbred lines from different heterotic groups are used in the crossing program [8,9]. Thus, knowledge on the molecular diversity, genetic purity, heterotic grouping, and genetic relationships among inbred lines developed by different research institutions are important for identifying the right parental combinations for initiating new crosses either for inbred line development or hybrid development. In crosses with known heterotic groups, inbred lines are derived by crossing two genetically distant lines with acceptable phenotypic traits from within the same heterotic group. For hybrid development, on the other hand, parental lines should be genetically distant and from different heterotic groups.

Maize breeders develop numerous inbred lines based on phenotypic selection and assign them into different heterotic groups based on combining ability tests, such as diallel and line-by-tester analyses. Numerous molecular characterization studies have been conducted on maize using diverse types of molecular markers, sample size, and marker density. In recent years, single nucleotide polymorphism (SNPs) based on genotyping by sequencing (GBS) technology [10] has become very popular for multiple reasons, which have been reviewed in other studies [11,12]. This includes the use of GBS to study maize germplasm from international research centers [13,14,15,16], Europe [17], and China [8], but the method has not yet been used to investigate maize germplasm developed or adapted in India. Previous molecular characterization studies in maize grown in India primarily used simple sequence repeat (SSR) markers [18,19,20,21,22]. Although SSRs are highly polymorphic, multi-allelic, and codominant, they are not suitable for large-scale germplasm characterization due to high genotyping cost per sample, low throughput, a limited number of markers, high genotyping errors, and uneven distribution in the genome [23,24].

High-density SNPs via GBS are providing good opportunities to the researchers for low-cost and rapid turnaround time to quickly genotype their germplasm for different purposes, including genetic diversity assessment, mapping genes and quantitative trait loci (QTL) using bi-parental populations and genome-wide association mapping (GWAS) panels, and developing improved varieties using marker-assisted and genomic selection [25,26,27]. Linkage disequilibrium (LD) is a non-random association between alleles at various loci [28]. It is very useful in GWAS to determine the suitability of molecular marker density to identify genes and QTLs associated with traits of economic importance. The objectives of the present study were, therefore, to understand the molecular diversity, genetic purity, relatedness, population structure, and LD in maize inbred lines developed by different research centers in India and compare them with representative inbred lines from international research centers using high-density SNPs.

## 2. Results

### 2.1. SNP Discovery and Their Distribution

We used a total of 60,227 polymorphic SNPs, each with a minimum allele frequency of 0.05 and a maximum missing data of 20%, to understand the genetic diversity, population structure, and LD across 384 inbred lines originated from CIMMYT (62) and 17 research institutions/centers in India (322) (Appendix A). The polymorphic SNPs were distributed across all ten maize chromosomes with per chromosome number ranging from 2167 on Chr-10 to 8976 on Chr-1 (Table 1). The number of polymorphic SNPs decreased to 45,548 when they were filtered using a MAF of 0.10. The 60,227 polymorphic SNPs represented 20,085 loci, which is equivalent to three SNPs per locus. The number of loci varied from 1080 on Chr-9 to 2906 on Chr-1 (Table 1) with an average of 2009 loci per chromosome. The minor allele frequency (MAF) (Figure 1) shows a skew distribution with 51% of SNPs having MAF between 0.05 and 0.15 and only 17.0% between 0.30 and 0.50.

The 60,227 polymorphic SNPs cover 2057.3 Mb of the maize genome with each chromosome ranging in length from 110.77 Mb (Chr-9) to 307.0 Mb (Chr-1) with an average of 205.73 Mb per chromosome (Figure 2). The total physical distance covered during sequencing accounted for 89.4% of the total B73 genome (2300 Mb) [29]. The polymorphic SNPs provided the whole genome-wide coverage with a SNP at a mean distance of 34.20 kb (Figure 2). The average physical distance of SNPs on each chromosome ranged from 29.78 kb on Chr-4 to 69.67 kb on Chr-10.

As shown in Table 2, the mean nucleotide diversity, frequency of polymorphic sites per bp, and proportion of heterozygous loci observed in the data set were 0.28, 0.02, and 0.17, respectively. The average polymorphic information content (PIC) for 60,277 SNPs was 0.23. DQL 574-2, DQL 609-5, DQL 790-2-4 Analysis of various mutational changes in the sequenced data revealed that the percentage of transition mutations was much higher (72.56%) than that of the transversion (27.44%). The frequency of SNP mutation C/T was highest (0.36), whereas C/G was lowest (0.05). The genome-wide mean transition to transversions ratio was 2.64. Overall, the frequency of C/T and A/G transitions was almost similar (~0.36) in the genome, whereas among transversions, the frequency of G/T was highest (0.076), while C/G was the lowest (0.05). In the case of transversion mutations, the frequencies of A/T, A/C, G/T were near similar (0.073 to 0.076), except the C/G (0.05) (Table 2).

### 2.2. Genetic Relationship and Population Structure

Most of the pair-wise relative kinship values (58.5%) were equal or close to 0 and only 19.3% of the pairs were above 0.05 (Figure 3A), indicating a distant relationship and little redundancy on allele compositions among most of the inbred lines. Identity-by-state (IBS)-based genetic distance between pair of inbred lines varied from 0.07 to 0.36 with an overall average of 0.28 (Appendix A). Most pairs of lines (69.0%) differed by 20–30% (Figure 3B), followed by 30–40% of the scored alleles (29.3% of the pairs). Only 1.7% of the pairs of lines differed by <20% of the scored alleles, which suggests greater genetic variation even among sister lines that shared common parentage. However, some of the inbred lines such as DQL 609-5, CM 133, and CML 220-1 were found highly heterogenous (up to 43.8%),with an overall average of 17% for all 384 lines (Appendix A, Figure 4).

The neighbor-joining phylogenetic trees revealed four major clusters that consisted of 90, 89, 129, and 76 lines (Figure 5A, Appendix A). The 62 CIMMYT lines were assigned into the first group (8 lines), second group (4 lines), third group (45), and fourth group (5 lines). A plot of the first three axes from principal component analysis (PCA) revealed three major groups corresponding to the second group, fourth group, and a mix of both the first and third groups (Figure 5B).To understand the patterns of relationship among inbred lines in the phylogenetic trees and PCA, we constructed multiple plots using seven categorical variables (group membership from STRUCTURE, germplasm origin, types of corn, kernel color, maturity group, plant height group, and ear height placement). We did not find clear patterns by any of the categorical variables (Appendix A). However, the grouping was observed partially based on the source germplasm from where the lines were derived.

The model-based population structure analysis conducted using 60,227 SNPs showed the highest ΔK peak at K = 6 (Appendix A), which suggests the presence of six sub-populations (Figure 6, Appendix A). Inbred lines with a membership probability of ≥0.50 were assigned into the same group, while those with <0.50 were designated as a mixed group. Of the 384 inbred lines, 227 of them (59.1%) were assigned into one of the six groups and the remaining 155 lines (40.36%) were assigned into a mixed group. Each group consisted of between 8 and 128 inbred lines. The first group (G1) consisted of 9 lines that originated from ICAR-IIMR Ludhiana of which eight were sister lines derived from a cross DML 51 × DML 4-2-2 (Appendix A). Similarly, G3 and G5 consisted of 8 and 17 lines originating from ICAR-IIMR Ludhiana, which were derived primarily from Brasil 117 × ESM 113 and AMH 3436 respectively. G2 consisted of22 lines originating from six centers, while G4 had 45 lines from seven centers, including lines derived from Brasil 117 × ESM 113 (8 lines) and CML 269 × HKI 4 C 4B-1 (7 lines). G6 consisted of 128 lines originating from 18 centers of which 64 were developed by the ICAR-IIMR Ludhiana center. Half of the CIMMYT lines were assigned into a mixed group and the remaining were assigned into G2 (4 lines), G4 (3 lines), and G6 (24 lines).

### 2.3. Linkage Disequilibrium and Haplotype Blocks Distribution

LD (*r*^2^) values computed from 60,277 polymorphic SNPs ranged from 0 to 1. The distribution pattern of LD values between pairs of SNPs ranging from 0.2 to 1.0 is given in Figure 7. The pairwise mean values of *r*^2^ decreased rapidly with increasing physical distance (Figure 8). The decline in mean LD values was observed at a physical distance of ≥10.0 kb. The chromosome-wise mean *r*^2^ values computed across the genome at various physical distance intervals have been given in Appendix A. The overall mean value of LD at <0.1 kb was 0.69, which declined to 0.49 at 10.0–100.0 kb interval. It further declined to 0.40, 0.32 and 0.14 at 100–1000 kb, 1000–5000 kb and >100,000 kb, respectively. The chromosome wise mean LD values at various distance intervals ranged from *r*^2^ = 0.14 (at distance >100,000 kb on Chr-9) to 0.75 (0.2–0.3 kb on Chr-10). The highest mean LD was observed on Chr-3 (0.52) and minimum on Chr-2 (0.42). The overall mean LD value across the genome was 0.46.

Haplotype block distribution analyses based on LD estimated across the 384 maize lines genotyped with 60,227 SNP detected a total of 10,200 conserved haplotype blocks spanning 2057.32 Mb across all chromosomes. Among total haplotype blocks, 79.4% were of size ranges from 0 to 1 kb, 2.70% were of >100 kb and the remaining were 1 kb and 100 kb. Chromosome-wise haplotype blocks ranged from 285 on Chr-10 to 1556 on Chr-1, with an average of 1020 blocks per chromosome (Table 3). The total length of all haplotype blocks per chromosome varied from 880 (Chr-10) to 11,470 kb (Chr-3), with an average of 6898 kb. The 2.7% of the total haplotypes blocks across the chromosomes were extending from more than 100 kb size with a range of three haplotypes on Chr-10 to 45 on Chr-3. The total length of haplotypes with more than 100 kb size in different chromosomes varied from 473.3 kb on Chr-10 to 6800.9 on Chr-3 with an average of 4016.4 kb across the chromosomes. The maximum haplotype block length was observed on Chr-8 (199.99 kb) and the minimum on Chr-10 (180.03 kb).

## 3. Discussion

### 3.1. Genetic Diversity and Population Structure

Overall, the polymorphic SNPs used in the present study were uniformly distributed across the maize genome, which provides a better estimate of diversity in the panel. The average physical distance between pairs of SNPs across the genome was 34.2 kb, which indicates good coverage of the whole genome. The high polymorphic information content, frequency of polymorphic site per base pair, and mean nucleotide diversity all suggest the presence of high genetic variation in the panel [30,31,32]. The minor allele frequency (MAF) (Figure 1) showed a skew distribution with 51% of loci displaying a MAF between 0.05 and 0.15, which is expected in germplasm that have gone through intensive artificial selection. Selection introduces new sources of genetic variation that increase fitness under stress conditions, but the selected individuals or loci have very low to low initial frequencies [10].

Relative kinship is important to understand the extent of relatedness between pairs of lines with kinship values close to 0 among pairs of unrelated lines, <0.25 for half-sib, 0.5–1.0 for full-sib, and >1.0 highly similar lines [33]. In the present study, the pairwise relative kinship values for 58.5% of the inbred lines were either equal or close to 0, which suggests a lower level of relatedness and lack of redundancy in the genomic compositions of most inbred lines (Figure 3A). Our results are similar to Shu et al. [8] who also reported >66% of pairs of maize inbred lines that showed a kinship coefficient near zero. The lack of redundancy in the genomic composition among inbred lines is also evident from the high genetic distance observed among pairs of most inbred lines (Figure 3B, Appendix A).

Genetic purity (homogeneity) in inbred lines is very important in maize breeding because it directly affects the ability of the line to be used as a parent in new line development and hybrid formation. In the current study, overall average of 17% heterogeneity was observed in the panel which is relatively high. It emphasizes the purification of heterogeneous lines either by ear to row selection or re-selection from the original seeds lots of the line. Relatively normal field corn genotypes of ICAR-IIMR Ludhiana and CIMMYT were showing low heterogeneity as compared to the lines from the centers and quality protein maize lines of the IIMR.Most of the IIMR quality protein lines are still in the advanced stage of development (S_4_–S_6_ generations). However, the percent heterogeneity of CIMMYT lines was higher than reported before [34]. We suspected either pollen contamination or seed admixture during seed maintenance breeding.Our results on the high proportion of heterogeneity agree with a previous study by Ertiro [15], who studied 265 maize inbred lines from CIMMYT, the Ethiopian Institute of Agricultural Research (EIAR), and the International Institute of Tropical Agriculture (IITA) using 220,878 polymorphic SNPs. The authors considered about 22% of the 265 inbred lines as genetically pure (homogenous); the remaining 27% and 51% of the lines had a heterogeneity ranging from 5.1 to 12.4% and from 12.5 to 31.5%, respectively.

Our analysis using phylogenetic analysis (Figure 5A), PCA (Figure 5B), and the model-based population structure (Figure 6) suggest the presence of three to six possible groups. However, we did not find clear clustering patterns by the centers that developed the lines, type of corn, grain characteristics, and agronomic traits (maturity groups, plant height groups, etc.). Earlier Boakyewaa et al. [30] and Xie et al. [35] reported three groups each, and Vigouroux et al. [36] reported four in a set of 94, 90, and 350 maize genotypes, respectively. Most CIMMYT lines tend to cluster together, but there was no obvious differentiation between lines developed by CIMMYT and those by the National programs in India, which is expected due to the continual exchange of germplasm among breeders. As expected, however, inbred lines derived from the same cross or related source germplasm (pedigree) tend to cluster together.

### 3.2. Linkage Disequilibrium and Distribution of Haplotype Blocks

Population size and the number of markers play a significant role in LD values: the lesser the numbers of markers the larger would be the LD values [37]. Our interest here is to understand whether the 60,227 SNPs and the 384 inbred lines are optimal for GWAS. We observed a good range of LD (0 to 1) across the entire maize genome. The overall mean values of LD (*r*^2^) reported in the literature at different physical intervals varied from 0.14 to 0.75, which may be due to differences in population size and the number of markers [37,38,39]. In the present study, we observed a rapidly declining trend in mean LD with an increase in the physical distance at 10 kb or greater in the genome, which agrees with various previous studies in maize [37,38,39]. Since maize is a highly cross-pollinated and diverse species, a rapid decay in LD is expected. In the self-pollinated crops, however, it can prolong to a longer distance due to their minimum outcrossing rate [33,40].

The identification of haplotype blocks in the genome can further improve the power of QTL detection in an association mapping panel. Haplotype blocks of long-length sequences that possess more polymorphic SNPs tend to harvest large diversity among the genotypes. In this study, we observed different sizes and numbers of haplotype blocks across the chromosomes that consisted of variable numbers of SNPs. On Chr-3, for example, we observed the highest total blocks length and relatively more numbers of blocks and SNPs. Simultaneously, the overall mean LD value across the physical distances was also highest on this chromosome. LD, haplotype blocks, SNPs numbers, and diversity in the genome are interrelated and are important for GWAS study. Rather than studying the whole genome, it is advisable to first examine for a preliminary association using a panel carrying more haplotype blocks. Once a specific region gets detected, then small haplotype blocks in the same regions can be explored further for fine mapping. These findings on LD and haplotype blocks suggest that the current panel of 384 inbred lines has sufficient genetic variability for all parameters and therefore can be considered for GWAS.

## 4. Materials and Methods

### 4.1. Plant Material and DNA Isolation

The present study used an association mapping panel of 384 diverse tropical maize inbred lines from CIMMYT (62), and seventeen maize research institutes/centers in India (322). The panel consisted of different types of corns, *viz.*, normal field corn (281), quality protein maize (95), popcorn (6), and sweet corn (2) with different kernel colors (yellow, white, orange, and dark orange), maturity groups (early, medium and late), plant height (short, medium and long), and ear placement (low, medium and high) (Appendix A). The highest number of samples was 260 lines (183 field corn, 74 QPM, and 3 popcorn lines) from ICAR-IIMR, Ludhiana. Genomic DNA was extracted from 2–3 gm fresh leaf tissues using Gen Elute Plant Genomic DNA Miniprep Kit (Sigma Aldrich, St. Louis, MO, USA). Each inbred line was represented by a bulk of five randomly selected individuals. The DNA was quantified with Qubit (Thermo fFsher Inc., Waltham, MA, USA) and was normalized to 100 ng/uL for GBS library preparation, Illumina genome sequencing, and SNPs discovery.

### 4.2. Library Preparation and Genotyping

GBS Libraries were prepared by digesting 0.5 μg genomic DNA of each of the lines with *ApeKI* (New England Biolabs, Ipswitch, MA, USA) restriction endonuclease enzyme as described in previous studies [8,10]. Briefly, each sample was digested for 4 h at 75 °C followed by ligation with barcoded adapters having the complementarily sticky ends to the digested DNA. The Qubit^®^ 2.0 fluorometer was used to determine the concentration of the library, which was then diluted to a concentration of 1 ng/uL. Libraries with an appropriate insert of 328 bases and an effective concentration of more than 2 nM were used for sequencing. The sequencing was done using Illumina TrueSeq v3.0 pair-end sequencing with read lengths of 151 bp on HiSeq×10 Platform. Ninety-five samples (plus a blank negative control) were sequenced in one lane. The reversible terminator-based method was followed that enables the detection of a single base when they were incorporated in growing DNA strands.

The sequencing generated a total of ~17.58 million raw reads, which were saved into FASTQ files.Quality control steps were followed to filter the raw data set by removing reads with a quality score of <20reads, by checking for the perfectly matched barcode with the expected four base remnants of the enzyme cut site, minor allele frequencies. Reads with minimum quality control (Q) score of 20 were considered for further analysis. These reads were sorted and de-multiplexed according to their barcode. The filtered sequences from each genotype were aligned to the maize reference genome using the Burrows-Wheeler alignment tool (BWA) [41] for SNPs identification. Subsequent processing, such as duplicate removal was performed using PICARD, (https://sourceforge.net/projects/picard/files/picard-tools/1.48/, accessed on 8 March 2022), [42]. FreeBayes [43] was used to identify genetic variants. The raw SNPs were then filtered using VCFtools using a maximum number of alleles of 2 and minimum coverage of 3. The SNPs were finally filtered using a minor allele frequency of 0.05 and missing data of ≤20% missing data were retained for further analysis, which resulted in a total of 60,227 polymorphic SNPs for all subsequent data analyses.

### 4.3. Data Analysis

The population structure was determined using the model-based STRUCTURE v.2.3.4 software (https://web.stanford.edu/group/pritchardlab/structure_software/release_versions/v2.3.4/html/structure.html, accessed on 8 March 2022) using K values ranging from 2 to 9, three independent runs at 150,000 Markov Chain Monte Carlo (MCMC) iterations, and 100,000 burn-in periods. Genotypes with membership probability values of ≥0.50 were assigned to the same group, while those with ≤0.50 probability were treated as mixed [44,45]. The optimal number of sub-populations in the panel was decided using the ΔK curve with maximum likelihood value (*ln Pr (X|K)*) for the run. Relative kinship and identity by state-based genetic distance between a pair of inbred lines, and principal component analysis (PCA) were done using TASSEL v5.2.80 (https://www.maizegenetics.net/tassel, accessed on 8 March 2022). The genetic distance matrix was used to construct a neighbor-joining phylogenetic tree in TASSEL. The Newick file from TASSEL was visualized using interactive Tree Of Life (iTOL) [46] (https://itol.embl.de/, accessed on 8 March 2022). The first three principal components from PCA were plotted using R platform. We used seven categorical variables in the PCA and phylogenetic trees, which included the research centers that developed the inbred lines, group membership from population structure analysis, type of corn, kernel color, maturity group, plant height, and ear height placement.

The linkage disequilibrium (LD) analysis was carried out using three data sets in Tomahawk software (https://mklarqvist.github.io/tomahawk/, accessed on 8 March 2022). The first data set consisted of all the 60,227 polymorphic SNPs, each with a minor allele frequency of 0.05. We then created two additional data sets by further filtering the SNPs using a minor allele frequency of 0.10 and 0.20. The LD values were computed by calculating pairwise squared allele-frequency correlations (*r*^2^) between the SNPs in the genome [47]. The LD values were estimated for different intervals ranging from 0.0–0.0001 Mb to 10–100 Mb on the various chromosomes. The SNPs positions in the genome were then used to study the LD decay for each chromosome and across the entire genome [48,49]. The distribution of haplotype blocks on various chromosomes was studied using the TASSEL v. 5.2 package (https://www.maizegenetics.net/tassel, accessed on 8 March 2022).

## 5. Conclusions

This study investigated the genetic diversity, relationship, population structure, and LD in a panel of 384 maize inbred lines using 60,277 polymorphic SNPs. Our study revealed the presence of large genetic variation among the inbred lines and no clear grouping among categorical variables. The inbred lines grouped partially on the basis of their pedigree. The presence of relatively high heterogeneous lines in the panel required special attention of the breeder for their purification depending on the purpose of the breeding program and the intended use of the inbred lines.The germplasm characterized in the current study is an important resource for breeding and genomics studies in India. The high-density genotype data and results presented in this study would be very useful for different purposes, including parental selection for new breeding programs, heterotic grouping, and genome-wide association studies.

## Figures and Tables

**Figure 1 plants-11-00799-f001:**
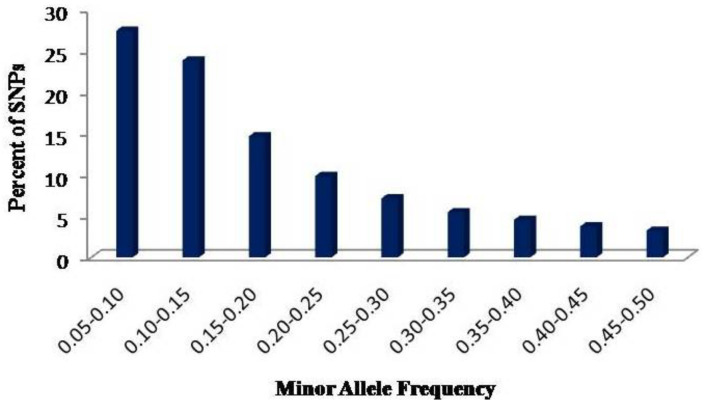
Frequency distribution of the 60,227 polymorphic SNPs by minor allele frequency.

**Figure 2 plants-11-00799-f002:**
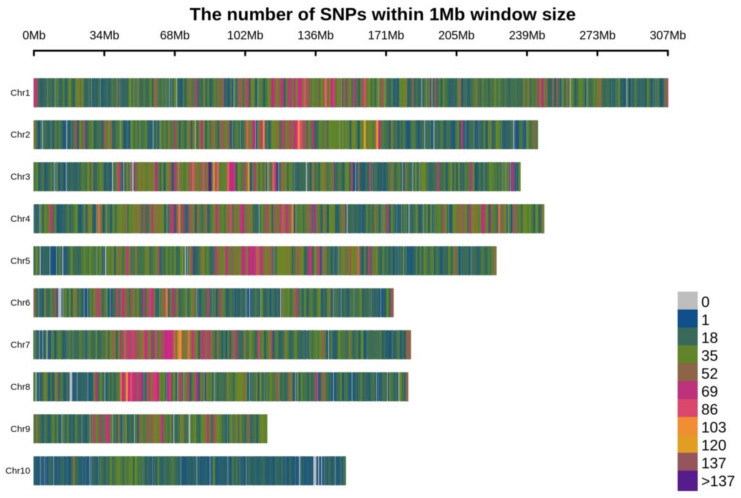
Distribution of SNPs used in the study on ten maize chromosomes. The horizontal axis shows the physical distance with the different colors reflecting the SNP density distribution.

**Figure 3 plants-11-00799-f003:**
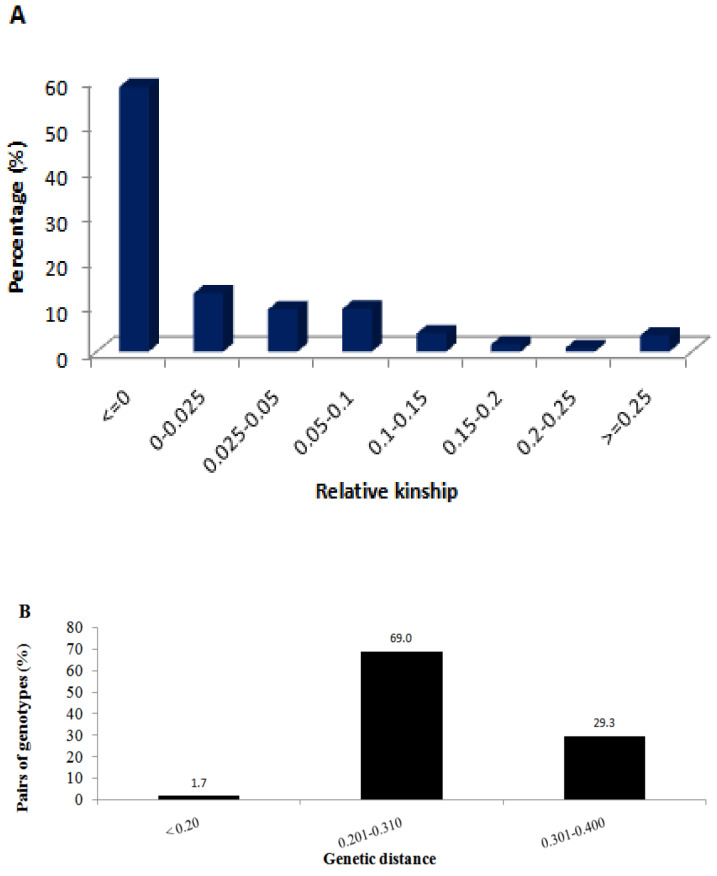
Summary of relative kinship and identity-by-state based genetic distance between pairs of the 384 inbred lines computed from 60,227 polymorphic SNPs. See Appendix A for genetic distance values and Appendix A for the pair-wise kinship between inbred lines.

**Figure 4 plants-11-00799-f004:**
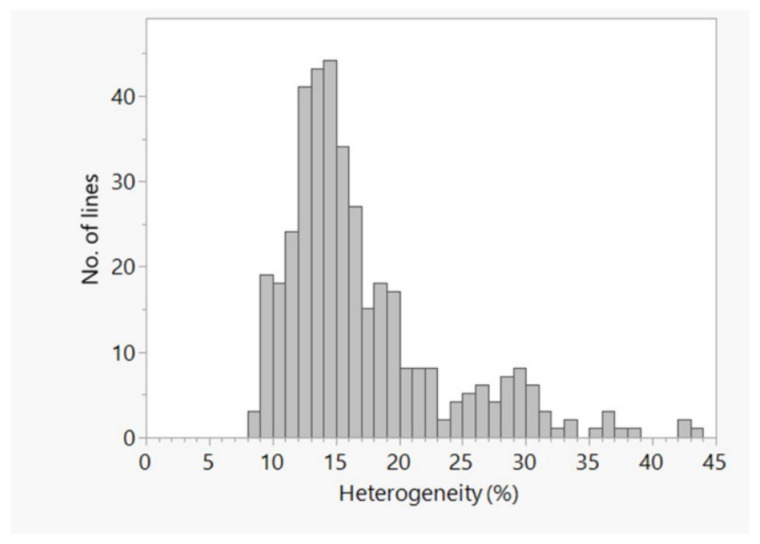
Frequency distribution plot of the percentage of heterogeneity within each of the 384 inbred lines based on 60,227 polymorphic SNPs.

**Figure 5 plants-11-00799-f005:**
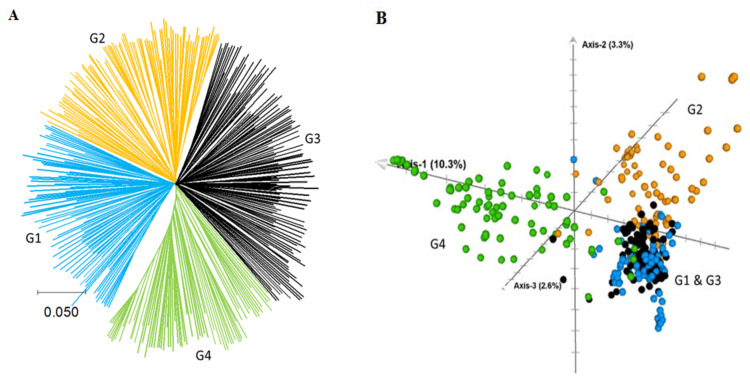
Neighbour-joining tree and multidimensional scaling (MDS) analysis of 384 maize inbred lines based on genetic distance matrix computed from 60,227 polymorphic SNP markers in TASSEL v 5.2. NJ tree (**A**) and MDS (**B**) were constructed using MEGA v11 and CurlyWhirly v1.21.08.16, respectively.

**Figure 6 plants-11-00799-f006:**
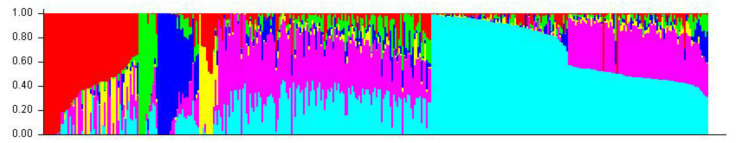
Population structure of 384 maize inbred lines based on 60,227 polymorphic SNPs at K = 6. Each inbred line is represented by a vertical line that is partitioned into K-colored segments, with lengths proportional to the estimated probability membership value (y-axis). The six groups are shown in different colors: G1 (Green), G2 (Pink), G3 (Yellow), G4 (Red), G5 (Blue), and G6 (Cyan). See Appendix A for details on group membership.

**Figure 7 plants-11-00799-f007:**
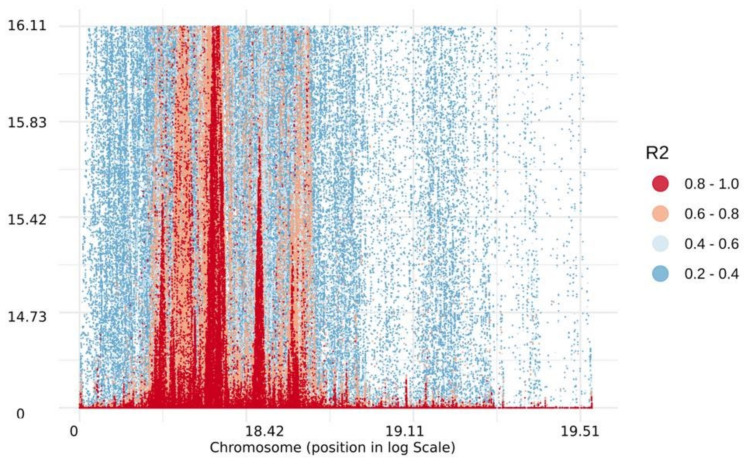
Distribution of linkage disequilibrium between individual SNPs pairs across the genome of 384 inbred lines genotyped with 60,277 SNPs. The horizontal and vertical axis representing the SNPs pairs and legend insert inside depicts the various classes of LD values ranging from 0.2 to 1.0.

**Figure 8 plants-11-00799-f008:**
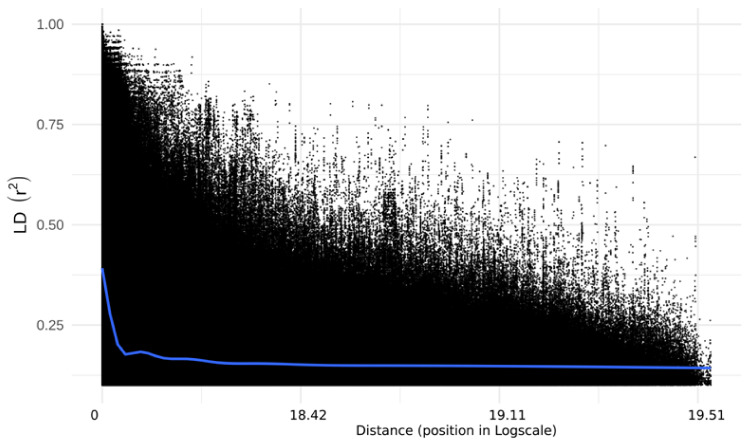
Linkage disequilibrium decay curve based on 60,227 polymorphic SNPs in 384 maize inbred lines. The blue line indicates the general trend in LD decay.

**Table 1 plants-11-00799-t001:** Chromosome-wise distribution of the polymorphic SNPs at different minor allele frequencies (MAF), physicalmapping length, and different loci based on 384 diverse maize inbred lines.

Chromosome (Chr.)	Physical Map Length (Mb)	Number of SNPs at MAF 0.05	Number of Loci	SNPs at MAF (≥0.1)
Chr-1	307.04	8976	2906	6946
Chr-2	244.44	7173	2299	5377
Chr-3	235.67	7405	2353	5757
Chr-4	246.99	8293	2676	6270
Chr-5	223.9	6823	2270	5142
Chr-6	174.03	4624	1511	3523
Chr-7	182.38	6053	1983	4456
Chr-8	181.12	5345	1733	4134
Chr-9	110.77	3368	1080	2618
Chr-10	150.98	2167	1274	1325
Total	2057.32	60,227	20,085	45,548

**Table 2 plants-11-00799-t002:** Summary of transition and transversion mutations detected among 384 maize genotypes.

Parameter	Value	Type of Mutation	SNP Mutation	Number of SNPs	Total SNPs per Type of Mutation
Number of SNPs	60,227	Transitions	A/G	21,775	43,701
Ts/Tv ratio	2.64		C/T	21,926	
Mean Nucleotide Diversity	0.28	Transversions	A/T	4435	16,526
			A/C	4505	
Frequency of polymorphic sites per bp	0.02		C/G	2996	
			G/T	4590	
Proportion of heterogenous SNPs	0.17		Total	60,227	60,227

**Table 3 plants-11-00799-t003:** The summary and distribution of haplotypes blocks on maize chromosomes.

Chr.	Chr. Length (Mb)	Hap. Freq (%)	^a^ Nohb	Total Block Length (kb)	^b^ Nohb	^c^ Sum (>100 kb)	^d^ TSNP	^e^ Max (kb)	^f^ Max SNPs	^g^ Min SNPs
Chr1	307.04	30.86	1556	9220	33	4894.75	5614 (334)	198.16	28	2
Chr2	244.44	30.01	1182	8740	37	5385.36	4378 (342)	199.69	27	2
Chr3	235.67	30.40	1282	11,470	45	6800.93	4784 (384)	199.28	26	2
Chr4	246.99	30.87	1432	9040	33	4843.49	5184 (272)	193.46	19	2
Chr5	223.90	31.08	1137	7160	29	3978.95	4198 (276)	196.90	22	2
Chr6	174.03	30.97	799	3410	11	1577.28	2767 (133)	193.12	25	2
Chr7	182.38	30.08	1065	9620	43	6333.56	3888 (387)	197.22	30	2
Chr8	181.12	30.89	876	6990	30	4317.02	3420 (346)	199.99	23	2
Chr9	110.77	30.35	586	3450	10	1559.55	2122 (96)	198.88	18	2
Chr10	150.98	36.35	285	880	3	473.303	744 (13)	180.03	8	2
Mean	205.73	31.19	1020	6998	27	4016.42	3710	195.67	23	2

^a^ The total number of haplotype blocks; ^b^ The number of haplotype blocks longer than 100 kb on each chromosome; ^c^ The total length of haplotype blocks longer than 100 kb for each chromosome; ^d^ The total number of SNPs in haplotype blocks and in longer than 100 kb blocks (in parenthesis); ^e^ Maximum size (in kb) of the haplotype block; ^f^ Max (SNPs) correspond to the maximum number of SNPs forming blocks; ^g^ Min (SNPs) correspond to the minimum number of SNPs forming blocks.

## Data Availability

The raw data is available with the first author for any further information/queries. Besides, part of it has been submitted with the manuscript as Appendix A.

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
