# Peer review of "Genetic Diversity, Population Structure and Linkage Disequilibrium Analyses in Tropical Maize Using Genotyping by Sequencing"

_plants, 2022, doi:10.3390/plants11060799_

Round 1
Reviewer 1 Report
Please have this manuscript proof read by a native English speaker.
Author Response
COMMENTS OF REVIEWER-1
Comment:
Please have this manuscript proof read by a native English speaker.
Response: As suggested by the reviewer, we have improved the language in the revised manuscript.
Reviewer 2 Report
This is more an academic manuscript with an average merit. However, given the effort put and amount of data used, I believe it is worth publishing with minor revision. I have few suggestions as follows:
- I do not see any statements about the objective of the study. Please include those towards the end of the introduction section
- Please remove Fig. 2 and Table 3 from the main body of this manuscript and put them under supplementary materials
- Fig. 5 [x-axis and y-axis] and Fig. 6 [x-axis]: convert the units into log scale. This is the standard practice in visualization of data with such a wide range
- Please include a section in discussion that explains the general characteristics of germplasm belonging to different clusters and/or population structures groups.
All the best.
Author Response
COMMENTS OF REVIEWER-2
This is more an academic manuscript with an average merit. However, given the effort put and amount of data used, I believe it is worth publishing with minor revision. I have few suggestions as follows:
Comment: I do not see any statements about the objective of the study. Please include those towards the end of the introduction section
Response: We are thankful to the reviewer for very constructive suggestions and for regarding the manuscript as “worth being published”. As suggested by the reviewer, we have written the objective of the study at the end of the Introduction in the revised manuscript.
Comment:
Please remove Fig. 2 and Table 3 from the main body of this manuscript and put them under supplementary materials.
Response: We have shifted Fig. 2 and Table 3 in supplementary material in the revised MS.
Comment:
Fig. 5 [x-axis and y-axis] and Fig. 6 [x-axis]: convert the units into log scale. This is the standard practice in visualization of data with such a wide range.
Response: As suggested by the reviewer, in the revised manuscript we have done the same. These figures are given as Fig. 6 and 7 in the revised version.
Comment:
Please include a section in discussion that explains the general characteristics of germplasm belonging to different clusters and/or population structures groups.
Response: The suggestions of the reviewer are well taken. We have added the text explaining explains the general characteristics of germplasm under the discussion section.

Reviewer 3 Report
- Statement The quantum of genetic gain by the use of conventional breeding approaches for the exploitation of heterosis has slowed down due to the lack of information on the genetic diversity of breeding lines provided at the Introduction deserves a more analytical explanation. Heterosis has actually not slowed down in the US (or perhaps too negligible for practical consideration), because interpopulation selection has not been neglected as in other countries, so, interpopulation improvement is the actual key in the long term, rather than information on genetic diversity. Please discuss this topic further.
- The first three lines of the Conclusion section correspond to Materials and Methods. Please remove them from the conclusion.
Author Response
COMMENTS OF REVIEWER-3
Comment: Statement The quantum of genetic gain by the use of conventional breeding approaches for the exploitation of heterosis has slowed down due to the lack of information on the genetic diversity of breeding lines provided at the Introduction deserves a more analytical explanation. Heterosis has actually not slowed down in the US (or perhaps too negligible for practical consideration), because interpopulation selection has not been neglected as in other countries, so, interpopulation improvement is the actual key in the long term, rather than information on genetic diversity. Please discuss this topic further.
Response: As suggested by the reviewer, in the revised manuscript we have discussed the same.
Comment: The first three lines of the Conclusion section correspond to Materials and Methods. Please remove them from the conclusion.
Response: As suggested by the reviewer, in the revised manuscript we have removed the same from the conclusion section.